# Boosting Discriminative Visual Representation Learning with Scenario-Agnostic Mixup

## Abstract

Mixup is a hit data-dependent augmentation technique that entails two sub-tasks: mixed sample generation and classification. This paper comprehensively studies the objective of mixup generation and proposes **S**cenario-**A**gnostic **Mix**up (SAMix) to address the two remaining challenges in this field at once:

**(i) Huge performance variation over scenarios caused by trivial solutions.** The objective of mixup generation narrows to selected sample pairs rather than the whole observed data manifold, which gives rise to the hassle of trivial solutions, resulting in drastic variations in sample mixing performance over different scenarios.

**(ii) Self-supervised learning (SSL) dilemma for online training policies.** While recent online training policies can generate out-of-manifold samples on supervised learning (SL), simply applying them to SSL scenarios leads to subpar performance.

We hypothesize and verify the objective function of mixup generation as optimizing *local smoothness* between two mixed classes subject to *global discrimination* from the other classes. Thus, we propose $\eta$-balanced mixup loss for complementary learning of the two sub-objectives. For the generation model, a label-free generator, Mixer, is designed to generate non-trivial mixed samples with great transferability. To reduce the computational cost from online training, we further introduce a pretrained version, SAMix$^{\mathcal{P}}$, which is more applicable and achieves more favorable generalizability. Extensive experiments on 12 SL and SSL image benchmarks show the consistent superiority of SAMix compared with state-of-the-art methods.

## 1 Introduction

By generating symmetric mixed data and labels, data mixing, or mixup, has significantly improved the generalization ability of deep neural networks (DNNs) in discriminative representation learning across a wide range of scenarios (Zhang et al., 2018; Kim et al., 2020; Lee et al., 2021). Despite its widespread application, the mixed sample generation policy necessitates an *explicit* hand-crafted design (*e.g.*, linear interpolation, or random local patch replacement). Instead, the *offline-optimizable* mixup strategies utilize labels to pinpoint task-relevant targets (*e.g.*, gradCAM (Selvaraju et al., 2019)) so as to generate semantically mixed samples in which, for instance, the saliency information from corresponding data can be maximized offline (Kim et al., 2020; Uddin et al., 2021; Kim et al., 2021). Zhu et al. (2020) learns a mixup generator by supervised adversarial training. These *hand-crafted* mixup methods are shown in the red box of Figure 2. Their performance varies greatly over scenarios.

As of late, it is a common practice to incorporate linear mixup methods with a contrastive learning paradigm (Kalantidis et al., 2020; Lee et al., 2021; Shen et al., 2021). More recently, AutoMix (Liu et al., 2022b) introduces a novel perspective to make the mixup framework parameterized and can be trained online. Although these *online-optimizable* methods have attained significant gains on supervised learning (SL) tasks, they still do not exploit the underlying structure of the whole observed data manifold, resulting in trivial solutions without the label guidance, which makes them fail to apply to self-supervised learning (SSL) scenarios (discussed in Section 2). The problem then naturally arises as to whether it is possible to design a more generalized and trained mixup policy that can be applied to both SL and SSL scenarios. To achieve this goal, there are two remaining challenges to be solved: **(i) how to keep the mixup performance stable over different scenarios by solving trivial solutions; (ii) how to make the online training policies generalizable for SSL scenarios.**

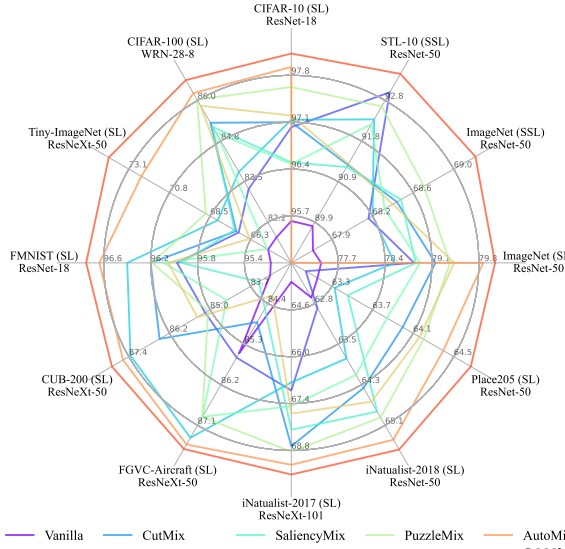

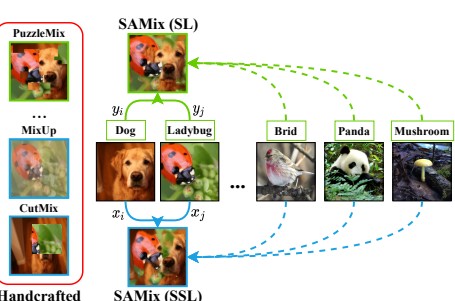

Figure 2: Illustration of SAMix, which exploits local and global information for mixup generation while exerting no dependency on labels. Labels are required for the mixed samples in green boxes but not for the blue ones. The solid line denotes that the local relationship influences the mixed sample directly, whereas the dashed one indicates that in-samples of the other classes serve as global constraints on the current mixed data.

Figure 1: Performance radar plots of mixup benchmark on a wide range of scenarios with ResNet variants, including 10 supervised learning (SL) datasets and 2 self-supervised (SSL) tasks on STL-10 and IN-1K.

In this paper, we propose **S**cenario-**A**gnostic **Mix**up (SAMix), a unified mixup framework as shown in Figure 2 that employs $\eta$-*balanced mixup loss* (in Section 3.1) for treating mixup generation and classification differently from local and global perspectives. At the same time, specially designed *Mixer* (in Section 3.2) generate mixed samples adaptively either at *instance-level* or *cluster-level* to tackle trivial solutions. Moreover, to further eliminate the risk of poor applicability and computational overhead in optimization, we propose a pre-trained version, SAMix$^{\mathcal{P}}$, which employs a pre-trained Mixer to generate mixed samples for balancing performance and speed for downstream applications. Surprisingly, SAMix$^{\mathcal{P}}$ can achieve competitive or slightly better performance than the online SAMix on classification tasks. Extensive experiments show the consistent superiority and generalizability of SAMix across 12 SL and SSL image benchmarks. Our contributions are summarized as follows:

- We first unravel the mixup learning objective into local and global terms and further analyze their corresponding properties (local smoothness and global discrimination) for mixup generation, then targeted propose $\eta$-balanced loss to boost the mixup generation quality.
- We propose a label-free mixed sample generator, Mixer, with mixing attention and non-linear content modeling which tackles the trivial solution problem effectively under the online optimizable framework and thus makes it more generalizable for a wide range of scenarios.
- Incorporating the label-free Mixer and $\eta$-balanced loss, a unified scenario-agnostic mixup training framework, SAMix, is proposed with consistent impressive performance that tackles the two problems and supports online and pre-trained pipelines for both SL and SSL tasks.
- Built upon SAMix framework, a pre-trained version SAMix$^{\mathcal{P}}$ is provided, which brings SAMix more favorable performance-efficiency trade-offs and better generalizability across multifarious visual downstream tasks.

## 2 PRELIMINARIES

Given a finite set of i.i.d samples, $X = [x_i]_{i=1}^n \in \mathbb{R}^{D \times n}$, each data $x_i \in \mathbb{R}^D$ is drawn from a mixture of, say $C$, distributions $\mathcal{D} = \{\mathcal{D}_c\}_{c=1}^C$. Our basic assumption for discriminative representations is that each component distribution $\mathcal{D}_c$ has relatively low-dimensional intrinsic structures, *i.e.,* the distribution $\mathcal{D}_c$ is constrained on a sub-manifold, say $\mathcal{M}_c$ with dimension $d_c \ll D$. The distribution $\mathcal{D}$ of $X$ is consisted of sub-manifolds, $\mathcal{M} = \cup_{c=1}^C \mathcal{M}_c$. We seek a low-dimensional representation $z_i \in \mathcal{M}$ of $x_i$ by learning a continuous mapping by a network encoder, $f_\theta(x) : x \longmapsto z$ with the parameter $\theta \in \Theta$, which captures intrinsic structures of $\mathcal{M}$ and facilitates the discriminative tasks.

Here we consider mixup as a generation task into discriminative representation learning to form a closed-loop framework. Then we have two mutually benefited sub-tasks: (a) *mixed data generation*

and (b) *classification*. As for the sub-task (a), we define two functions, $h(\cdot)$ and $v(\cdot)$, to generate mixed samples and labels with a mixing ratio $\lambda \sim Beta(\alpha, \alpha)$. Given the mixed data, (b) defines a mixup training objective to optimize the representation space between instances or classes.

**Mixup classification as the main task.**   Since we aim to seek a good representation to facilitate discriminative tasks, the mixed samples should be diverse and well-characterized. The mixed samples with semantic information can be easily obtained by parametric learning, while it becomes a challenge without supervision. Therefore, two types of mixup classification objective $\mathcal{L}_{\theta, \omega}$ can be defined for *class-level* and *instance-level* mixup training. As for parametric training, given two randomly selected data pairs $(x_i, y_i)$ and $(x_j, y_j)$, the mixed data is generated as $x_m = h(x_i, x_j, \lambda)$ and $y_m = v(y_i, y_j, \lambda)$. The objective of the *class-level* mixup is as,

$$\ell^{CE}(y_m, p_m) = \lambda \ell^{CE}(y_m, p_m) + (1 - \lambda)\ell^{CE}(y_m, p_m). \tag{1}$$

Notice that we fix $v(\cdot)$ as the linear interpolation in our discussions, *i.e.*, $v(y_i, y_j, \lambda) \triangleq \lambda y_i + (1-\lambda)y_j$. Symmetrically, we denote $h(\cdot)$ as a pixel-wise mixing policy with element-wise product $\odot$ for most input mixup methods (Zhang et al., 2018; Yun et al., 2019; Kim et al., 2020), *i.e.*, $x_m = s_i \odot x_i + s_j \odot x_j$, where $s_i \in \mathbb{R}^{H \times W}$ is a pixel-wise mask and $s_j = 1 - s_i$. Notice that each coordinate $s_{w,h} \in [0, 1]$. We can generate $x_m$ with a pair of randomly selected samples $(x_i, x_j)$ and formulate mixup infoNCE loss for *instance-level* mixup:

$$\ell^{NCE}(z_m) = \lambda \ell^{NCE}(z_m, z_i) + (1 - \lambda)\ell^{NCE}(z_m, z_j), \tag{2}$$

where $z_m$, $z_i$ and $z_j$ denote the corresponding representations. The major difference with Eq. 1 is that the augmentation view that generates $z_m$ is not from the same view, *i.e.*, $z_i$ and $z_j$, as the objective function, which is effective in retaining task-relevant information, details in A.5.1.

**Mixup generation as the auxiliary task.**   Unlike the learning object on the *unmixed* data $X$ in Sec. 2, the performance of (b) mixup classification mainly depends on the quality of (a) mixup generation. We thus regard (a) as an auxiliary task to (b) and model $h(\cdot)$ as a sub-network $\mathcal{M}_\phi$ with the parameter $\phi \in \Phi$, called Mixer. Specifically, (a) aims to generate a pixel-wise mask $s \in \mathbb{R}^{H \times W}$ for mixing sample pairs. The mixup mask $s_i$ should directly related to $\lambda$ and the contents of $(x_i, x_j)$. Practically, our $\mathcal{M}_\phi$ takes $l$-th layer feature maps $z^l \in \mathbb{R}^{C_l \times H_l \times W_l}$ and $\lambda$ value as the input, $\mathcal{M}_\phi : x_i, x_j, z_i^l, z_j^l, \lambda \longmapsto x_m$. The generation process of $\mathcal{M}_\phi$ can be trained by a mixup generation loss as $\mathcal{L}_\phi^{gen}$, and a mask loss designed for generated mask $s_i$ denoted as $\mathcal{L}_\phi^{mask}$. Formally, we have the mixup generation loss as $\mathcal{L}_\phi = \mathcal{L}_\phi^{cls} + \mathcal{L}_\phi^{mask}$, and the final learning objective is,

$$\min_{\theta, \omega, \phi} \mathcal{L}_{\theta, \omega} + \mathcal{L}_\phi. \tag{3}$$

Both $\mathcal{L}_{\theta, \omega}$ and $\mathcal{L}_\phi$ can be optimized alternatively in a unified framework using a momentum pipeline with the stop-gradient operation (Grill et al., 2020; Liu et al., 2022b), as shown in Figure 5(b) (left). **In SSL, however, this framework can easily make the generator fall into a trivial solution.** To solve this problem, we propose a novel mixup loss function and a generator architecture, Mixer. Combining the two can fully exploit the ability of mixup in learning discriminative features.

## 3   SAMix for Discriminative Representation Learning

### 3.1   Learning Objective for Mixup Generation

Typically the objective function $\mathcal{L}_\phi$ corresponding to the mixup generation is consistent with the classification in parametric training (*e.g.*, $\ell^{CE}$). In this work, we argue that mixup generation is aimed to *optimize the local term subject to the global term*. The local term centers on the classes of sample pairs to be mixed, while the global term introduces the constraints of other classes. For example, $\ell^{CE}$ is the global term whose each class produces an equivalent effect on the final prediction without focusing on the relevant classes of the current sample pair. At the class level, to emphasize the local term, we introduce a parametric binary cross-entropy (pBCE) loss for the generation task. Formally, assuming $y_i$ and $y_j$ belong to the class $a$ and class $b$, pBCE can be summarized as:

$$\ell_+^{CE}(p_m) = -\lambda y_{i,a} \log p_m - (1 - \lambda)y_{j,b} \log p_m, \tag{4}$$

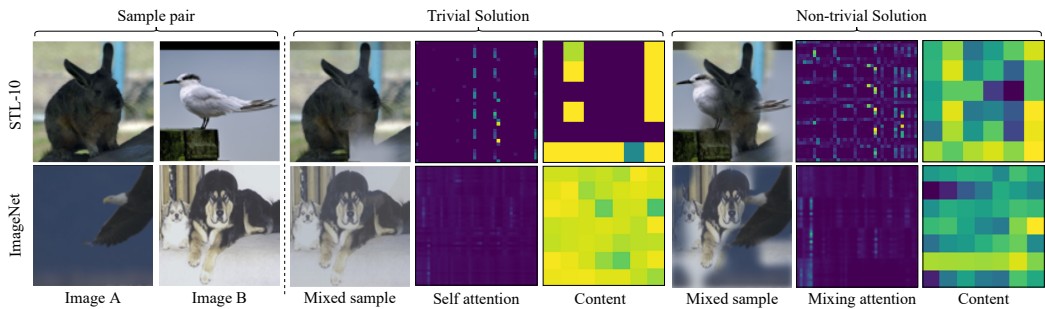

Figure 4: Visualization of trivial (using linear $\mathcal{C}$ and self-attention) and non-trivial solutions (using the proposed non-linear $\mathcal{C}_{NC}$ and mixing attention) for SAMix-I on STL-10 and IN-1k datasets.

where $y_{i,a} = 1$ and $y_{i,b} = 1$ denote the one-hot label for class $a$ and $b$. Notice that we use $\ell_+$ and $\ell_-$ to represent the local and global terms, and $\ell_-$ for the parametric loss refers to $\ell^{CE}$. Symmetrically, we have a non-parametric binary cross-entropy mixup loss (BCE) for CL:

$$\ell_+^{NCE}(z_m) = -\lambda \log p_{m,i} - (1-\lambda) \log p_{m,j}, \tag{5}$$

where $p_{m,i} = \frac{\exp(z_m z_i/t)}{\exp(z_m z_i/t) + \exp(z_m z_j/t)}$ and its $\ell_-$ refers to $\ell^{NCE}$.

**Balancing local and global terms.** Given that both local and global terms can contribute to mixup generation, we conduct empirical analysis on the importance of each term in both the SL and SSL scenarios to build a more balanced learning objective. We first analyze the properties of both terms with two hypothesizes: (i) the local term $\ell_+$ *determines* the generation performance, (ii) the global term $\ell_-$

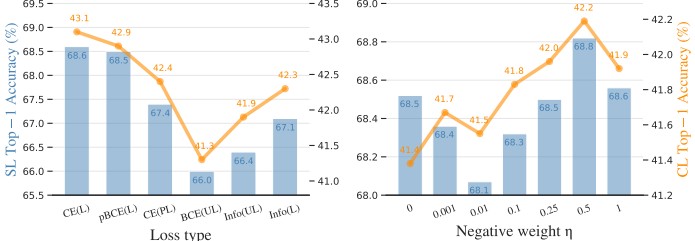

Figure 3: Analysis of the learning objective of Mixer on Tiny ImageNet with ResNet-18. **Left**: the results of various losses on the SL task (left y axis) and the CL task (right y axis). **Right**: the effect of using various negative weights $\eta$.

improves global discrimination but is sensitive to class information. To verify these properties, we design an empirical experiment based on the proposed Mixer on Tiny (see A.5). The main difference between the mixup CE and infoNCE is whether to adopt parametric class centroids. Therefore, we compare the intensity of class information among unlabeled (UL), pseudo labels (PL), and ground truth labels (L). Notice that PL is generated by ODC (Zhan et al., 2020) with the cluster number $C$. The class supervision can be imported to mixup infoNCE loss by filtering out negative samples with PL or L as (Khosla et al., 2020) denoted as infoNCE (L) and infoNCE (PL). As shown in Figure 3 (left), our hypothesizes are verified in the SL task (as the performance decreases from CE(L) to pBCE(L) and CE(PL) losses), but the opposite result appears in the CL task. The performance increases from InfoNCE(UL) to InfoNCE(L) as the false negative samples are removed (Robinson et al., 2021; Khosla et al., 2020) while trivial solutions occur using BCE(UL) (in Figure 6). Therefore, we propose it is better to explicitly import class information as PL for instance-level mixup to generate "strong" inter-class mixed samples while preserving intra-class compactness.

**SAMix with $\eta$-balanced generation objectives.** In practice, we provide two versions of learning objective: the mixup CE loss with PL as the class-level version (SAMix-C), and the mixup infoNCE loss as the instance-level one (SAMix-I). Then, we hypothesize that the best performing mixed samples should be close to the sweet spot: *achieving $\lambda$ local smoothness between two classes or neighborhood systems while globally discriminating from the other classes or instances.* Built upon this view, we specially design an $\eta$-balanced mixup loss as the objective of mixup generation,

$$\ell_\eta = \ell_+ + \eta \ell_-, \ \eta \in [0,1]. \tag{6}$$

As shown in Figure 3 (right), we empirically analyze the performance of employing various $\eta$ in Eq. 6 on Tiny and find out that using $\eta = 0.5$ performs the best on both the SL and CL tasks. In the end, we provide the learning objective, $\mathcal{L}_\phi^{cls} \triangleq \ell_+^{CE} + \eta \ell_-^{CE}$, with L for class-level mixup and with PL for SAMix-C, $\mathcal{L}_\phi^{cls} \triangleq \ell_+^{NCE} + \eta \ell_-^{NCE}$ for SAMix-I (more details in A.2).

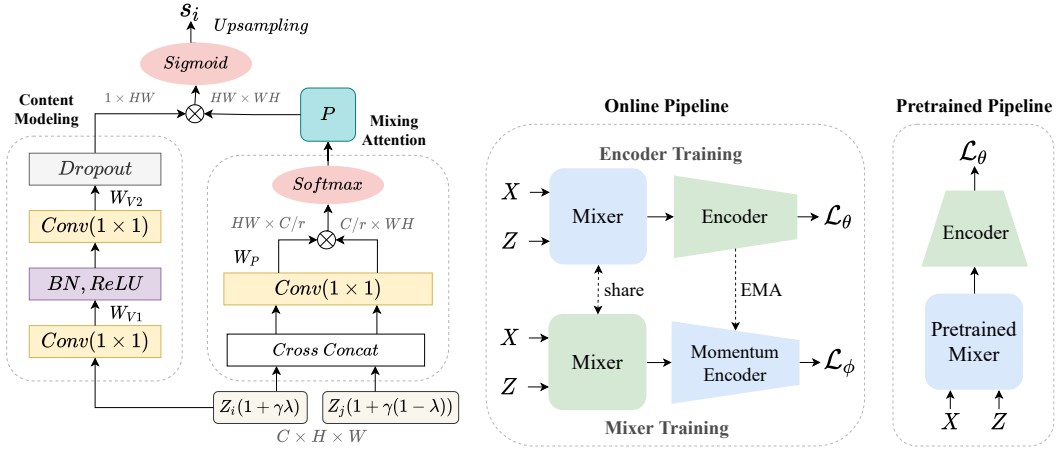

(a) Mixer network architecture      (b) Two training pipelines of Mixer

Figure 5: (a) The network architecture of the proposed Mixer for mixup generation. (b) Framework comparison of the popular AutoMix online pipeline (left) and our pre-trained pipeline (right). $X$ and $Z$ denote the input and corresponding feature maps from the Momentum Encoder. Blue modules are not updated by back-propagation. The online pipeline optimizes Mixer and Encoder alternatively, while our pre-trained one adopts pre-trained Mixer on large datasets.

## 3.2 DE NOVO MIXER FOR MIXUP GENERATION

While AutoMix introduces the online MixBlock to adaptively learn mixup generation policy, there are still three key limitations in practice: **(a) fail to encode the mixing ratio $\lambda$ on small datasets; (b) trivial solutions when performing CL tasks; (c) the online training pipeline leads to far more computational costs than MixUp.** Our Mixer $\mathcal{M}_\phi$ solves these issues on an individual basis.

**Adaptive $\lambda$ encoding and mixing attention.** Since a randomly sampled $\lambda$ should directly guide mixup generation, the predicted mask $s$ should be semantically proportional to $\lambda$. Nevertheless, previous work regards $\lambda$ as the prior knowledge and concatenates $\lambda$ to input feature maps, which might fail to encode $\lambda$ properly (detailed analysis in A.6.1). We propose an *adaptive $\lambda$ encoding* as,

$$z_{i,\lambda}^l = (1 + \gamma\lambda)z_i^l, \tag{7}$$

where $\gamma$ is a learnable scalar constrained to $[0, 1]$. Notice that $\gamma$ is initialized to 0 during training. As shown in Figure 5(a), we compute the mixing relationship between two samples by a new *mixing attention*: we concatenate $(z_{i,\lambda}^l, z_{j,1-\lambda}^l)$ as the input, $\tilde{z}^l = \text{concat}(z_{i,\lambda}^l, z_{j,1-\lambda}^l)$, and compute the attention matrix as,

$$P_{i,j} = \psi\left(\frac{(W_P\tilde{z}^l)^T \otimes W_P\tilde{z}^l}{\mathcal{N}(\tilde{z}^l)}\right), \tag{8}$$

where $\psi(\cdot)$ is the softmax function, $\mathcal{N}(\tilde{z}^l)$ denotes a normalization factor, and $\otimes$ is matrix multiplication. Notice that the mixing attention provides both the cross-attention between $z_{i,\lambda}^l$ and $z_{j,\lambda}^l$ and the self-attention of each feature itself.

**Non-linear content modeling.** In vanilla self-attention mechanism, the content sub-module $\mathcal{C}$ is a linear projection, $C_i = W_z\tilde{z}^l$, where $W_z$ denotes a $1 \times 1$ convolution. However, we find the training process of this structure is unstable when performing CL tasks with the linear $\mathcal{C}$ in the early period and sometimes trapped in trivial solutions, such as all coordinates on $s_i$ predicted as a constant. As shown in Figure 4, we visualize $C_i$ and $P_{i,j}$ of trivial and non-trivial results and find that the trivial $s_i$ is usually caused by a constant $C_i$. We hypothesize that trivial solutions happen earlier in the linear $\mathcal{C}$ than in $P_{i,j}$, because it might be unstable to project high-dimensional features to 1-dim linearly. Hence, we design a *non-linear content modeling* sub-module $\mathcal{C}_{NC}$ that contains two $1 \times 1$ convolution layers with a batch normalization layer and a ReLU layer in between, as shown in Figure 5(a). To increase the robustness and randomness of mixup training, we add a Dropout layer with a dropout ratio of 0.1 in $\mathcal{C}_{NC}$. Formally, Mixer $\mathcal{M}_\phi$ can be written as,

$$s_i = U\left(\sigma\left(\psi\left(\frac{(W_P\tilde{z}^l)^T \otimes W_P\tilde{z}^l}{\mathcal{N}(\tilde{z}^l)}\right) \otimes \mathcal{C}_{NC}(z_{i,\lambda}^l)\right)\right). \tag{9}$$

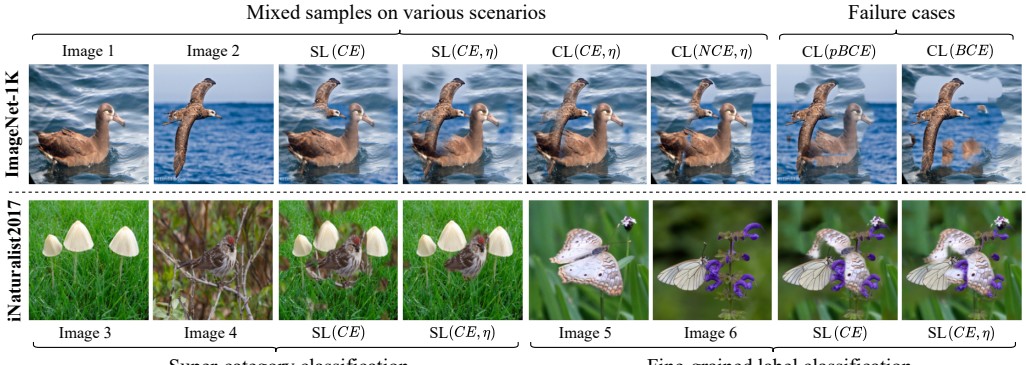

Figure 6: Visualization of the mixed samples from Mixer within various learning scenarios on IN-1k and iNat2017. Note that $\lambda = 0.5$ and $\eta = 0.5$ if the balance coefficient $\eta$ is included. CL(C) and CL(I) denote using SAMix-C and SAMix-I, respectively.

**Pre-trained pipeline *v.s.* Online pipeline.** Even though current online-optimized mixup methods (Liu et al., 2022b) outperform their handcrafted counterparts by a substantial margin, the computational cost is intolerable, especially for large datasets. On large-scale benchmarks, it is empirically evident that the samples generated by SAMix in both early and late training phases or with different CNN encoders vary little. Thus, we hypothesize that, akin to knowledge distillation (Dabouei et al., 2021), **the online training Mixer can be replaced with a pre-trained one.** From this view, we design a pre-trained SAMix pipeline, denoted as $\text{SAMix}^{\mathcal{P}}$, as shown in Figure 5(b). Extensive experiments lead us to the following conclusions: (i) $\text{SAMix}^{\mathcal{P}}$ pre-trained on large datasets achieves comparative or slightly better performance than the online SAMix on relevant datasets with lower computational cost. (ii) $\text{SAMix}^{\mathcal{P}}$ with lightweight CNN encoders (*e.g.*, ResNet-18) yields better performance than the heavy ones (*e.g.*, ResNet-101). (iii) $\text{SAMix}^{\mathcal{P}}$ exhibits better transferabilities than the popular $\text{AutoMix}^{\mathcal{P}}$. (iv) The online training pipeline is still irreplaceable on small-scale datasets (*e.g.*, CIFAR and CUB).

**Prior knowledge of mixup.** We summarize the commonly adopted prior knowledge (Kim et al., 2020; Dabouei et al., 2021) for mixup as two aspects: (a) adjusting the mean of $s_i$ correlated with $\lambda$, and (b) balancing the smoothness of local image patches while maintaining discrimination of $x_m$. Based on them, we introduce $\lambda$ *adjusting* and modifying the mask loss $\mathcal{L}_\phi^{mask}$ (details in A.2).

## 3.3 EMPIRICAL ANALYSIS AND DISCUSSION

To showcase the impact of local and global constraints on mixup generation, we visualize mixed samples generated by Mixer on various scenarios with different types of class clustering distributions, *i.e.*, uniform general classification scenarios on IN-1K, and imbalanced fine-grained scenarios on iNat2017. In Figure 6, It is evident that SAMix captures robust underlying data structure from both class- and instance-level effectively and thus can sidestep trivial solutions and then increase its applicability for SSL.

**Class-level.** Within the supervised tasks, global constraint localizes significant features by discriminating them from the other classes, while the local term is apt to preserve more information tied to the two given samples and classes. For instance, comparing the mixed results with and without $\eta$-balanced mixup loss, it was found that pixels of the foreground target were of interest to Mixer. When the global constraint is balanced ($\eta = 0.5$), the foreground target is retained more completely. Notably, the Mixer we proposed remains **invariant to the background for the more challenging fine-grained classification while at the same time preserving discriminative features**.

**Instance-level.** Since no label is available for SSL, the global and local terms are transformed from class to instance level. Similar results are shown in the top row of Figure 6, and the only difference is that SAMix-C exhibits a more precise target correspondence compared to SAMix-I via introducing class information by PL, which further indicates the importance of the information of classes. If we only focus on local relationships, Mixer can only generate mixed samples with fixed patterns as the last two results in the top row of Figure 6. These failures imply the importance of global constraints.

# 4 EXPERIMENTS

We first evaluate SAMix for supervised learning (SL) in Sec. 4.1 and self-supervised learning (SSL) in Sec. 4.2, and then perform ablation studies in Sec. 4.3. Nine benchmarks are used for evaluation: CIFAR-100 (Krizhevsky et al., 2009), Tiny-ImageNet (Tiny) (Chrabaszcz et al., 2017), ImageNet-1k (IN-1k) (Russakovsky et al., 2015), STL-10 (Coates et al., 2011), CUB-200 (Wah et al., 2011), FGVC-Aircraft (Aircraft) (Maji et al., 2013), iNaturalist2017/2018 (iNat2017/2018) (Horn et al., 2018), and Place205 (Zhou et al., 2014). All experiments are conducted with PyTorch and reported the *mean of 3 trials*. SAMix uses $\alpha = 2$ and the feature layer $l = 3$ while SAMix$^{\mathcal{P}}$ is pre-trained 100 epochs with ResNet-18 (SL tasks) or ResNet-50 (SSL tasks) on IN-1k. A momentum training coefficient for SAMix is increased from 0.999 to 1 in a cosine curve. The *median* validation top-1 accuracy of the last 10 epochs is recorded.

## 4.1 EVALUATION ON SUPERVISED IMAGE CLASSIFICATION

CNNs and ViTs are used as backbone networks, including ResNet (R), Wide-ResNet (WRN) (Zagoruyko & Komodakis, 2016), ResNeXt-32x4d (RX) (Xie et al., 2017), DeiT (Touvron et al., 2021), and Swin Transformer (Liu et al., 2021). We employ PyTorch training procedures Paszke et al. (2019) by default: an SGD optimizer with cosine scheduler Loshchilov & Hutter (2016). A special case in Table 4: RSB A3 (using LAMB optimizer (You et al., 2020) for R-50) in timm (Wightman et al., 2021) and DeiT (using AdamW optimizer (Loshchilov & Hutter, 2019) for DeiT-S and Swin-T) training recipes are fully adopted on IN-1k. MCE and MBCE denote mixup cross-entropy and mixup binary cross-entropy in RSB A3. For a fair comparison, grid search is performed for hyper-parameters $\alpha \in \{0.1, 0.2, 0.5, 1, 2, 4\}$ of all mixup variants. We follow hyper-parameters in original papers by default. $*$ denotes *arXiv* preprint works, † and ‡ denote reproduced results by official codes and originally reported results, the rest are reproduced (see A.3 and A.4).

**Comparison and discussion** Table 3 illustrates the results on small-scale and fine-grained classification tasks. SAMix consistently improves classification performance over the previous best algorithm, AutoMix, with the improved Mixer. Notably, SAMix significantly improved the mixup performance on CUB-200 and Aircraft by 1.24% and 0.78% based on ResNet-18, and continued to expand its dominance on Tiny by bringing 1.23% and 1.40% improvement on ResNet-18 and ResNeXt-50. As for the large-scale classification task, we benchmark popular mixup methods in Table 1, 2, and 4, SAMix and SAMix$^{\mathcal{P}}$ outperform all existing methods on IN-1k, iNat2017/2018 and Places205. Surprisingly, SAMix$^{\mathcal{P}}$ yields comparable or even better performance than SAMix with lower computational cost.

Table 1: Top-1 Acc (%) of image classification on IN-1k training 100-epoch and 300-epoch using procedures.

| Methods | PyTorch 100ep | | | | | PyTorch 300ep | | | |
| | R-18 | R-34 | R-50 | R-101 | RX-101 | R-18 | R-34 | R-50 | R-101 |
|---|---|---|---|---|---|---|---|---|---|
| Vanilla | 70.04 | 73.85 | 76.83 | 78.18 | 78.71 | 71.83 | 75.29 | 77.35 | 78.91 |
| Mixup | 69.98 | 73.97 | 77.12 | 78.97 | 79.98 | 71.72 | 75.73 | 78.44 | 80.60 |
| CutMix | 68.95 | 73.58 | 77.17 | 78.96 | 80.42 | 71.01 | 75.16 | 78.69 | 80.59 |
| ManifoldMix | 69.98 | 73.98 | 77.01 | 79.02 | 79.93 | 71.73 | 75.44 | 78.21 | 80.64 |
| SaliencyMix | 69.16 | 73.56 | 77.14 | 79.32 | 80.27 | 70.21 | 75.01 | 78.46 | 80.45 |
| FMix* | 69.96 | 74.08 | 77.19 | 79.09 | 80.06 | 70.30 | 75.12 | 78.51 | 80.20 |
| PuzzleMix | 70.12 | 74.26 | 77.54 | 79.43 | 80.63 | 71.64 | 75.84 | 78.86 | 80.67 |
| ResizeMix* | 69.50 | 73.88 | 77.42 | 79.27 | 80.55 | 71.32 | 75.64 | 78.91 | 80.52 |
| AutoMix | 70.50 | 74.52 | 77.91 | 79.87 | 80.89 | 72.05 | 76.10 | 79.25 | 80.98 |
| **SAMix**$^{\mathcal{P}}$ | 70.83 | 74.95 | 78.06 | **80.05** | 80.98 | 72.27 | 76.28 | 79.39 | **81.10** |
| **SAMix** | 70.85 | 74.96 | 78.11 | 80.02 | 81.03 | 72.33 | 76.35 | 79.40 | 81.06 |

Table 2: Top-1 Acc (%) of image classification on iNat2017/2018 and Places205.

| Method | iNat2017 | | iNat2018 | | Places205 | |
| | R-50 | RX-101 | R-50 | RX-101 | R-18 | R-50 |
|---|---|---|---|---|---|---|
| Vanilla | 60.23 | 63.70 | 62.53 | 66.94 | 59.63 | 63.10 |
| Mixup | 61.22 | 66.27 | 62.69 | 67.56 | 59.33 | 63.01 |
| CutMix | 62.34 | 67.59 | 63.91 | 69.75 | 59.21 | 63.75 |
| ManifoldMix | 61.47 | 66.08 | 63.46 | 69.30 | 59.46 | 63.23 |
| SaliencyMix | 62.51 | 67.20 | 64.27 | 70.01 | 59.50 | 63.33 |
| FMix* | 61.90 | 66.64 | 63.71 | 69.46 | 59.51 | 63.63 |
| PuzzleMix | 62.66 | 67.72 | 64.36 | 70.12 | 59.62 | 63.91 |
| ResizeMix* | 62.29 | 66.82 | 64.12 | 69.30 | 59.66 | 63.88 |
| AutoMix* | 63.08 | 68.03 | 64.73 | 70.49 | 59.74 | 64.06 |
| **SAMix**$^{\mathcal{P}}$ | **63.38** | 68.23 | **65.16** | 70.56 | 59.82 | **64.35** |
| **SAMix** | 63.32 | 68.26 | 64.84 | 70.54 | 59.86 | 64.27 |

Table 3: Top-1 Acc (%) of supervised image classification on CIFAR-100, Tiny-ImageNet, CUB-200 and Aircraft.

| Method | CIFAR-100 | | | Tiny-ImageNet | | CUB-200 | | FGVC-Aircraft | |
| | R-18 | RX-50 | WRN-28-8 | R-18 | RX-50 | R-18 | RX-50 | R-18 | RX-50 |
|---|---|---|---|---|---|---|---|---|---|
| Vanilla | 78.04 | 81.09 | 81.63 | 61.68 | 65.04 | 77.68 | 83.01 | 80.23 | 85.10 |
| Mixup | 79.12 | 82.10 | 82.82 | 63.86 | 66.36 | 78.39 | 84.58 | 79.52 | 85.18 |
| CutMix | 78.17 | 81.67 | 84.45 | 65.53 | 66.47 | 78.40 | 85.68 | 78.84 | 84.55 |
| ManifoldMix | 80.35 | 82.88 | 83.24 | 64.15 | 67.30 | 79.76 | 86.38 | 80.68 | 86.60 |
| SaliencyMix | 79.12 | 81.53 | 84.35 | 64.60 | 66.55 | 77.95 | 83.29 | 80.02 | 84.31 |
| FMix* | 79.69 | 81.90 | 84.21 | 63.47 | 65.08 | 77.28 | 84.06 | 79.36 | 84.85 |
| PuzzleMix | 80.43 | 82.57 | 85.02 | 65.81 | 66.92 | 78.63 | 84.51 | 80.76 | 86.23 |
| ResizeMix* | 80.01 | 81.82 | 84.87 | 63.74 | 65.87 | 78.50 | 84.77 | 78.10 | 84.08 |
| AutoMix* | 82.04 | 83.64 | 85.16 | 67.33 | 70.72 | 79.87 | 86.56 | 81.37 | 86.69 |
| **SAMix** | 82.30 | 84.42 | 85.50 | 68.89 | 72.18 | 81.11 | 86.83 | 82.15 | 86.80 |

Table 4: Top-1 Acc (%) of image classification on IN-1k.

| Methods | R-50 (A3) | | DeiT-S | Swin-T |
| | MCE | MBCE | MCE | MCE |
|---|---|---|---|---|
| Mixup+CutMix | 76.49 | 78.08 | 79.80 | 81.20 |
| Mixup | 76.01 | 77.66 | 79.65 | 81.01 |
| CutMix | 76.47 | 77.62 | 79.78 | 81.23 |
| AttentiveMix | 76.78 | 77.46 | 80.32 | 81.29 |
| SaliencyMix | 76.85 | 77.93 | 79.32 | 81.37 |
| PuzzleMix | 77.27 | 78.02 | 79.84 | 81.47 |
| TransMix‡ | - | - | 80.70 | 80.80 |
| TokenMix‡ | - | - | 80.80 | 80.60 |
| AutoMix* | 77.45 | 78.34 | 80.75 | 80.80 |
| **SAMix**$^{\mathcal{P}}$ | 77.80 | **78.73** | 80.87 | 80.80 |
| **SAMix** | 78.33 | 78.45 | 80.94 | 81.87 |

Table 5: Top-1 Acc (%) of linear classification for pre-trained models on STL-10.

| CL method | Method | R18 400ep | R18 800ep | R50 400ep | R50 800ep |
|---|---|---|---|---|---|
| MoCo.V2 | - | 81.50 | 85.64 | 84.89 | 89.68 |
| | Mixup | 84.51 | 87.93 | 88.24 | 92.20 |
| | ManifoldMix | 84.17 | 87.70 | 88.06 | 91.65 |
| | CutMix | 84.28 | 87.60 | 87.51 | 90.81 |
| MoCo.V2 | SaliencyMix | 84.33 | 87.27 | 87.35 | 90.77 |
| | FMix* | 84.43 | 87.68 | 88.14 | 91.56 |
| | ResizeMix* | 83.88 | 87.25 | 86.88 | 90.83 |
| MoCo.V2 | **SAMix-I** | **85.44** | **88.58** | **88.87** | **92.41** |
| SwAV† (C) | - | 81.10 | 85.56 | 84.35 | 88.79 |
| MoCo.V2(C) | Inter-Intra* | 84.89 | 87.85 | 88.33 | 92.24 |
| MoCo.V2(C) | PuzzleMix* | 84.98 | 88.07 | 88.40 | 91.98 |
| **MoCo.V2(C)** | **SAMix-C** | **85.60** | **88.63** | **88.91** | **92.45** |

Table 6: Top-1 Acc (%) of linear classification pre-trained on Tiny-ImageNet and ImageNet-1k.

| CL method | Method | Tiny R18 | Tiny R50 | IN-1k R18 | IN-1k R50 |
|---|---|---|---|---|---|
| MoCo.V2 | - | 38.29 | 42.08 | 52.85 | 67.66 |
| MoCo.V2 | Mixup | 41.24 | 46.61 | 53.03 | 68.07 |
| MoCo.V2 | CutMix | 41.62 | 46.24 | 52.98 | 68.28 |
| MoCo.V2 | SaliencyMix | 41.14 | 46.13 | 53.06 | 68.31 |
| MoCo.V2(C) | PuzzleMix* | 41.86 | 46.72 | 53.46 | 68.48 |
| MoCHi† | Mixup+latent | 41.78 | 46.55 | 53.12 | 68.01 |
| i-Mix† | Mixup+latent | 41.61 | 46.57 | 53.09 | 68.10 |
| UnMix‡ | Mixup+latent | - | - | - | 68.60 |
| WBSIM‡ | Mixup+CutMix | - | - | - | 68.40 |
| **MoCo.V2** | **SAMix-I** | 41.97 | 47.23 | 53.75 | 68.76 |
| **MoCo.V2** | **SAMix-I$^{\mathcal{P}}$** | **43.57** | **48.10** | 53.72 | 68.82 |
| **MoCo.V2(C)** | **SAMix-C** | **43.68** | 47.51 | **53.93** | **68.86** |

Table 7: Top-1 Acc (%) of linear classification of ResNet-50 pre-trained with various SSL methods on IN-1k.

| Method | SimCLR | MoCo.V1 | BYOL | SwAV | SimSiam | MoCo.V3 |
|---|---|---|---|---|---|---|
| PT Epoch | 200 | 200 | 300 | 200 | 200 | 300 |
| - | 61.6 | 61.0 | 72.3 | 69.1 | 70.0 | 72.8 |
| Mixup | 61.6 | 61.2 | 72.4 | 69.2 | 70.1 | 72.8 |
| CutMix | 61.8 | 61.5 | 72.5 | 69.5 | 70.3 | 73.0 |
| i-Mix†(2-Mix) | 61.7 | 61.4 | - | 69.4 | - | 72.8 |
| SDMP†(3-Mix) | 62.3 | 61.7 | - | - | - | **73.5** |
| **SAMix-I$^{\mathcal{P}}$** | 62.2 | 61.7 | 72.6 | 69.8 | 70.4 | 73.2 |
| **SAMix-C$^{\mathcal{P}}$** | **62.4** | **61.9** | **72.8** | **69.9** | **70.5** | 73.4 |

Table 8: Top-1 Acc (%) of linear classification of ViT-S pre-trained on IN-1k.

| Method | MoCo.V2 | BYOL | SwAV | MoCo.V3 |
|---|---|---|---|---|
| PT Epoch | 300 | 300 | 300 | 300 |
| - | 72.7 | 71.4 | 73.5 | 73.2 |
| CutMix | 72.6 | 71.2 | 73.6 | 73.0 |
| i-Mix†(2-Mix) | 71.8 | - | 73.3 | 72.7 |
| DACL†(1-Mix) | 72.5 | - | - | 72.9 |
| SDMP†(3-Mix) | **72.9** | - | - | **73.8** |
| **SAMix-I$^{\mathcal{P}}$** | 72.8 | 72.7 | 73.6 | 73.4 |
| **SAMix-C$^{\mathcal{P}}$** | **72.9** | **72.8** | **73.8** | 73.6 |

## 4.2 EVALUATION ON SELF-SUPERVISED LEARNING

Then, we evaluate SAMix on various SSL tasks pre-training on STL-10, Tiny, and IN-1k. We adopt all hyper-parameter configurations from MoCo.V2 unless otherwise stated. We compared SAMix in three dimensions in CL: (i) compare with other mixup variants, based on our proposed cross-view pipeline, and whether the predefined cluster information is given (denotes by C) or not, as shown in Table 5. (ii) longitudinal comparison with CL methods that utilize mixup variants or Mixup+*latent* space mixup strategies in Table 6, including MoCHi (Kalantidis et al., 2020), i-Mix (Lee et al., 2021), Un-Mix (Shen et al., 2021), and WBSIM (Chu et al., 2022), where all comparing methods are based on MoCo.V2 except SwAV (Caron et al., 2020). (iii) extend SAMix$^{\mathcal{P}}$ and mixup variants to various CL baselines based on ResNet-50 and ViT-S in Table 7 and Table 8, compared with DACL (Verma et al., 2021) and SDMP (Ren et al., 2022) (using three mixup augmentations). In these tables, ⋆ denotes our modified methods (PuzzleMix* uses PL and Inter-Intra* combines inter-class CutMix with intra-class Mixup, and $n$-Mix denotes the types of mixup variants used in the SSL method.

**Linear Classification** Following the linear classification protocol proposed in MoCo, we train a linear classifier on top of frozen backbone features with the supervised train set. We train 100 epochs using SGD with a batch size of 256. The initialized learning rate is set to $0.1$ for Tiny and STL-10 while 30 for IN-1k, and decay by $0.1$ at epochs 30 and 60. As shown in Table 5, SAMix-I outperforms all the linear mixup methods by a large margin, while SAMix-C surpasses the saliency-based PuzzleMix when PL is available. And SAMix-I has both global and local properties through infoNCE and BCE losses. Meanwhile, Table 6 demonstrates that both SAMix-I and SAMix-C surpass other CL methods combined with the predefined mixup. Overall, SAMix-C yields the best performance in CL tasks, indicating it provides task-relevant information with the help of PL. Table 7 and Table 8 verify the generalizability of SAMix$^{\mathcal{P}}$ variants on popular CL baselines, which achieve comparable performances to recently proposed algorithms that combine $n$-Mix with CL.

**Downstream Tasks** Following the protocol in MoCo, we evaluate transferable abilities of the learned representation of comparing methods to object detection task on PASCAL VOC (Everingham et al., 2010) and COCO (Lin et al., 2014) in Detectron2 (Wu et al., 2019). We fine-tune Faster R-CNN (Ren et al., 2015) with pre-trained models on VOC *trainval07+12* and evaluate on the VOC *test2007* set. Similarly, Mask R-CNN (He et al., 2017) is fine-tuned ($2\times$ schedule) on the COCO *train2017* and evaluated on the COCO *val2017*. SAMix still achieves comparable performance among state-of-the-art mixup methods for CL. Please refer to A.5.5 for more results.

Figure 7: (a) Hyper-parameter $\alpha$ for mixup. (b) The cluster number $C$ for SAMix-C in CL tasks on Tiny. (c) Top-1 accuracy *v.s.* training time on IN-1k based on ResNet-50 with 100 epochs. (d) Top-1 accuracy *v.s.* training time *v.s.* GPU memory (G) on CIFAR-100 based on DeiT-S with 300 epochs.

## 4.3 ABLATION STUDY

We conduct ablation studies in four aspects: (i) **Mixer**: Table 10 verifies the effectiveness of each proposed module in both SL and CL tasks on Tiny. The first three modules enable Mixer to model non-linear mixup relationships, while the next two modules enhance Mixer, especially in CL tasks. (ii) **Learning objectives**: We analyze the effectiveness of proposed $\ell_\eta$ with other losses, as shown in Table 9. Using $\ell_\eta$ for the mixup CE and infoNCE consistently improves the performance both for the CL task on STL-10 and Tiny. (iii) **Time complexity analysis**: Figure 7 (c) shows computational analysis conducted on the SL task on IN-1k using PyTorch 100-epoch settings. Notice that the overall accuracy *v.s.* time efficiency of both SAMix and SAMix$^{\mathcal{P}}$ are superior to other methods. (iv) **Hyper-parameters**: Figure 7 (a) and (b) show ablation results of the hyper-parameter $\alpha$ and the clustering number $C$ for SAMix-C. We empirically select $\alpha$=2.0 and $C = 200$ as default.

Table 9: Ablation of learning objectives in the SSL tasks.

| Objective | STL-10 | Tiny |
|---|---|---|
| BCE | 85.25 | 41.28 |
| infoNCE | 85.36 | 41.85 |
| infoNCE ($\eta = 0.5$) | **85.44** | **41.97** |
| CE (PL) | 85.56 | 42.36 |
| CE (PL)+infoNCE | 85.41 | 42.12 |
| CE (PL, $\eta = 0.5$) | **85.60** | **42.53** |

Table 10: Ablation of modules in Mixer and the $\eta$-balanced loss.

| Designed module | SL | CL |
|---|---|---|
| Mixing attention | 67.17 | 40.58 |
| +Adaptive $\lambda$ | 67.95 | 41.82 |
| +Non-linear content | 68.46 | 42.45 |
| +$\mathcal{L}_{mask}$ | 68.57 | 42.68 |
| +$\lambda$ adjusting | 68.61 | 43.14 |
| +$\ell_\eta$ ($\eta = 0.5$) | **68.82** | **43.68** |

## 5 RELATED WORK

**Class-level Mixup for SL**    There are four types sample mixing policies for class-level mixup: linear mixup of input space (Zhang et al., 2018; Yun et al., 2019; Hendrycks et al., 2020; Harris et al., 2020; Qin et al., 2020) and latent space (Verma et al., 2019; Faramarzi et al., 2020), saliency-based (Uddin et al., 2021; Kim et al., 2020; 2021), generation-based (Zhu et al., 2020; Venkataramanan et al., 2022), and learning mixup generation and classification end-to-end (Liu et al., 2022b; Dabouei et al., 2021). More recently, mixup designed for ViTs optimizes mixing policies with self-attention maps (Chen et al., 2022; Liu et al., 2022a). SAMix belongs to the fourth type and learns both class- and instance-level mixup relationships, and its pre-trained SAMix$^{\mathcal{P}}$ eliminates high time-consuming problems of this type of method. Additionally, some researchers Park et al. (2022); Chen et al. (2022); Liu et al. (2023) improve class mixing policies upon linear mixup. Please refer to A.7 for details.

**Instance-level Mixup for SSL**    A complementary method to learn better instance-level representation is to apply mixup in SSL scenarios Lee et al. (2021). However, most existing approaches are limited to employing linear mixup variants, such as applying MixUp and CutMix in the input or latent space mixup (Kalantidis et al., 2020; Chu et al., 2022; Verma et al., 2021; Ren et al., 2022) for SSL without ground-truth labels. SAMix improves SSL performance by learning mixup policies online.

## 6 CONCLUSIONS AND LIMITATIONS

This paper study and decompose objectives for mixup generation as local-emphasized and global-constrained terms to adaptively learn a robust sample mixing policy at both class- and instance-level. SAMix provides a unified mixup framework with both online and pre-trained pipelines to boost discriminative representation learning based on improved $\eta$-balanced loss and Mixer. Moreover, a more applicable pre-trained version SAMix$^{\mathcal{P}}$ is provided. As a limitation, the Mixer only takes two samples as input and conflicts when the task-relevant information is overlapping. For future work, we suppose that k-mixup (k≥2) or conflict-aware Mixer can be the promising avenue to improve mixup.

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

## A  APPENDIX

The Appendix section is structured as follows:

(1.1) Appendix A.1 introduces dataset information used in the experiments.

(2.2) Appendix A.2 provides implementations of SAMix for supervised and contrastive learning.

(3.3) Appendix A.3 provides hyper-parameter settings for compared methods in image classification experiments.

(4.4) Appendix A.4 introduces implementation details and experiment settings of compared contrastive learning methods.

(5.5) Appendix A.5 provides experiment analysis and results for Sec. 3, including analysis of training pipeline and loss design for instance-level mixup, studying properties of inter- and intra-class instance mixup, Comparison of different objectives for mixup generation, and estimating mutual information between mixed and original samples.

(6.6) Appendix A.6 visualizes mixing attention, content modeling, the effect of losses, and comparison with existing methods.

(7.7) Appendix A.7 we further provide extensive related work of mixup augmentations and self-supervised learning.

### A.1  BASIC SETTINGS

**Reproduction details.**    We use OpenMixup (Li et al., 2022) implemented in PyTorch (Paszke et al., 2019) as our code-base for both supervised image classification and contrastive learning (CL) tasks. Except results marked by † and ‡, we reproduce most experiment results of compared methods, including Mixup (Zhang et al., 2018), CutMix (Yun et al., 2019), ManifoldMix (Verma et al., 2019), SaliencyMix (Uddin et al., 2021), FMix (Harris et al., 2020), and ResizeMix (Qin et al., 2020).

**Dataset information.**    We briefly introduce image datasets used in Sec. 4:  (1) CIFAR-100 (Krizhevsky et al., 2009) contains 50k training images and 10K test images of 100 classes. (2) ImageNet-1k (IN-1k) (Krizhevsky et al., 2012) contains 1.28 million training images and 50k validation images of 1000 classes. (3) Tiny-ImageNet (Tiny) (Chrabaszcz et al., 2017) is a rescaled version of ImageNet-1k, which has 100k training images and 10k validation images of 200 classes. (4) STL-10 (Coates et al., 2011) benchmark is designed for semi- or unsupervised learning, which consists of 5k labeled training images for 10 classes 100K unlabelled training images, and a test set of 8k images. (5) CUB-200-2011 (CUB) (Wah et al., 2011) contains over 11.8k images from 200 wild bird species for fine-grained classification. (6) FGVC-Aircraft (Aircraft) (Maji et al., 2013) contains 10k images of 100 classes of aircraft. (7) iNaturalist2017 (iNat2017) (Horn et al., 2018) is a large-scale fine-grained classification benchmark consisting of 579.2k images for training and 96k images for validation from over 5k different wild species.  (8) PASCAL VOC (Everingham et al., 2010) is a classical objection detection and segmentation dataset containing 16.5k images for 20 classes. (9) COCO (Lin et al., 2014) is an objection detection and segmentation benchmark containing 118k scenic images with many objects for 80 classes.

### A.2  IMPLEMENTATION OF SAMIX

**Online training pipeline.**    We provide the detailed implementation of SAMix in SL tasks. As shown in Figure 5(b) (left), we adopt the momentum pipeline (Grill et al., 2020; Liu et al., 2022b) to optimize $\mathcal{L}_{\theta,\omega}$ for mixup classification and $\mathcal{L}_\phi$ for mixup generation in Eq. 3 in an end-to-end manner:

$$\theta_q^t, \omega_q^t \leftarrow \underset{\theta,\omega}{\arg\min} \, \mathcal{L}_{\theta_q^{t-1}, \omega_q^{t-1}}, \tag{10}$$

$$\phi^t \leftarrow \underset{\phi}{\arg\min} \, \mathcal{L}_{\theta_k^t, \omega_k^t} + \mathcal{L}_{\phi^{t-1}}, \tag{11}$$

where $t$ is the iteration step, $\theta_q, \omega_q$ and $\theta_k, \omega_k$ denote the parameters of online and momentum networks, respectively. The parameters in the momentum networks are an exponential moving average of the online networks with a momentum decay coefficient $m$, taking $\theta_k$ as an example,

$$\theta_k^t \leftarrow m\theta_k^{t-1} + (1-m)\theta_q^t. \tag{12}$$

The training process of SAMix is summarized as four steps: (1) using the momentum encoder to generate the feature maps $Z^l$ for Mixer $\mathcal{M}_\phi$; (2) generating $X^q_{mix}$ and $X^k_{mix}$ by Mixer for the online networks and Mixer; (3) training the online networks by Eq. 10 and the Mixer by Eq. 11 separately; (4) updating the momentum networks by Eq. 12.

**Prior knowledge of mixup.** As we discussed in Sec. 3.2, we introduce some prior knowledge to the Mixer from two aspects: (a) To adjust the mean of $s_i$ correlated with $\lambda$, we introduce a mask loss that aligns the mean of $s_i$ to $\lambda$, $\ell_\mu = \beta \max(|\lambda - \mu_i| - \epsilon, 0)$, where $\mu_i = \frac{1}{HW} \sum_{h,w} s_{i,h,w}$ is the mean and $\epsilon = 0.1$ as a margin. Meanwhile, we propose a test time $\lambda$ *adjusting* method. Assuming $\mu_i < \lambda$, we adjust each coordinate on $s_i$ as $\hat{s}_i = \frac{\mu_i}{\lambda} s_i$, and $\hat{s}_j = 1 - \hat{s}_i$. (b) To balance the smoothness of local image patches and the discrimination (e.g., variance) of $x_m$, we adopt a bilinear upsampling as $U(\cdot)$ for smoother masks and propose a variance loss to encourage the sparsity of learned masks, $\ell_\sigma = \frac{1}{WH} \sum_{w,h} (\mu_i - s_{w,h})^2$. We summarize the mask loss as, $\mathcal{L}^{mask}_\phi = \beta(\ell_\mu + \ell_\sigma)$, where $\beta$ is a balancing weight. $\beta$ is initialized to $0.1$ and linearly decreases to $0$ during training.

## A.3 SUPERVISED IMAGE CLASSIFICATION

**Hyper-parameter settings.** As for hyper-parameters of SAMix, we follow the basic setting in AutoMix for both SL and SSL tasks: SAMix adopts $\alpha = 2$, the feature layer $l = 3$, the bilinear upsampling, and the weight $\beta = 0.1$ which linearly decays to 0. We use $\eta = 0.5$ for small-scale datasets (CIFAR-100, Tiny, CUB and Aircraft) and $\eta = 0.1$ for large-scale datasets (IN-1k and iNat2017). As for other methods, PuzzleMix (Kim et al., 2020), Co-Mixup (Kim et al., 2021), and AugMix (Hendrycks et al., 2020) are reproduced by their official implementations with $\alpha = 1, 2, 1$ for all datasets. As for mixup methods reproduced by us, we provide dataset-specific hyper-parameter settings as follows. For CIFAR-100, Mixup and ResizeMix use $\alpha = 1$, and CutMix, FMix and SaliencyMix use $\alpha = 0.2$, and ManifoldMix uses $\alpha = 2$. For Tiny, IN-1k, and iNat2017 datasets, ManifoldMix uses $\alpha = 0.2$, and the rest methods adopt $\alpha = 1$ for median and large backbones (e.g., ResNet-50). Specially, all these methods use $\alpha = 0.2$ (only) for ResNet-18. For small-scale fine-grained datasets (CUB-200 and Aircraft), SaliencyMix and FMix use $\alpha = 0.2$, and ManifoldMix uses $\alpha = 0.5$, while the rest use $\alpha = 1$.

## A.4 CONTRASTIVE LEARNING

**Implementation of SAMix-C and SAMix-I.** As for SSL tasks, we adopt the cross-view objective, $\ell^{NCE}(z_i^{\tau_q}, z_i^{\tau_k}) + \ell^{NCE}(z_m)$, where $z_i = z_i^{\tau_k}$ and $z_j = z_j^{\tau_k}$, for instance-level mixup classification in all methods (except for † and ‡ marked methods). We provide two variants, SAMix-C and SAMix-I, which use different learning objectives of mixup classification. The basic network structures (an encoder $f_\theta$ and a projector $g_\omega$) are adopted as MoCo.V2 (Chen et al., 2020b). Similar to SAMix in SL tasks, SAMix-C employs a parametric cluster classification head $g_\psi^C$ for online clustering (Caron et al., 2018; Zhan et al., 2020) to provide pseudo labels (PL) to calculate $\mathcal{L}_\phi^{cls}$. It takes feature vectors from the momentum encoder as the input (optimized by Eq. 11) and has no impact on the mixup classification objective for the online networks. Meanwhile, SAMix-I employs the instance-level classification loss for both $\mathcal{L}_{\theta,\omega}$ and $\mathcal{L}_\phi^{cls}$. Moreover, we use the proposed $\eta$-balanced mixup loss $\mathcal{L}_\phi^{cls}$ for both SAMix-C and SAMix-I with $\eta = 0.5$ and the objective $\mathcal{L}_\phi$ for Mixer.

**Hyper-parameter settings.** As for Table 5 and Table 6, all compared CL methods use MoCo.V2 pre-training settings except for SwAV (Caron et al., 2020), which adopts ResNet-50 (He et al., 2016) as the encoder $f_\theta$ with two-layer MLP projector $g_\omega$ and is optimized by SGD optimizer and Cosine scheduler with the initial learning rate of 0.03 and the batch size of 256. The length of the momentum dictionary is 65536 for IN-1k and 16384 for STL-10 and Tiny datasets. The data augmentation strategy is based on IN-1k in MoCo.v2 as follows: Geometric augmentation is `RandomResizedCrop` with the scale in $[0.2, 1.0]$ and `RandomHorizontalFlip`. Color augmentation is `ColorJitter` with {brightness, contrast, saturation, hue} strength of $\{0.4, 0.4, 0.4, 0.1\}$ with a probability of 0.8, and `RandomGrayscale` with a probability of 0.2. Blurring augmentation uses a square Gaussian kernel of size $23 \times 23$ with a std uniformly sampled in $[0.1, 2.0]$. We use $224 \times 224$ resolutions for IN-1k and $96 \times 96$ resolutions for STL-10 and Tiny datasets. As for Table 7 and Table 8, we follow the original setups of these CL baselines (SimCLR (Chen et al., 2020a), MoCo.V1 (He et al., 2020a), MoCo.V2 (Chen et al., 2020b), BYOL (Grill et al., 2020), SwAV (Caron et al., 2020), SimSiam (Chen

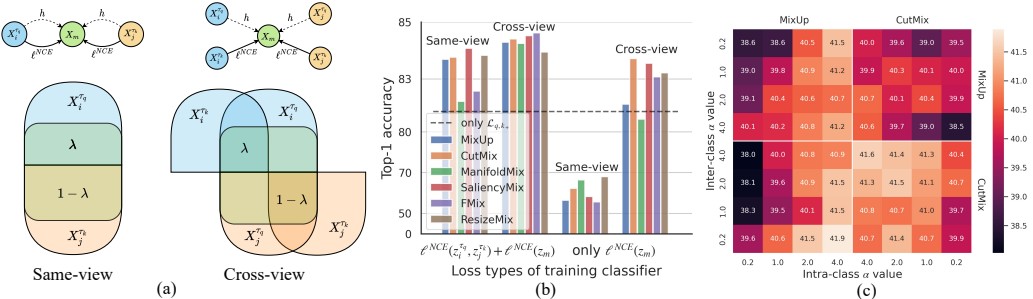

Figure 8: (a) Graphical models and information diagrams of *same-view* and *cross-view* training pipeline for instance-level mixup. Taking the *cross-view* as an example, $x_m = h(x_i^{T2}, x_j^{T2}, \lambda)$, the $\lambda$ region denotes the corresponding information partition for $\lambda I(z_m^{\tau_q}, z_i^{\tau_k})$ and the $1-\lambda$ region for $(1-\lambda)I(z_m^{\tau_q}, z_j^{\tau_k})$. (b) Linear evaluation (top-1 accuracy on STL-10) of whether to use the cross-view pipeline and to combine the original infoNCE loss with the mixup infoNCE loss. (c) A heat map of linear evaluation (top-1 accuracy on Tiny) represents the effects of using MixUp and CutMix as the inter-class (y-axis) and intra-class mixup (x-axis) using various $\alpha$.

& He, 2021), and MoCo.V3 (Chen et al., 2021)) using OpenMixup (Li et al., 2022) implementations. Notice that MoCo.V3 is specially designed for vision Transformers Dosovitskiy et al. (2021) while other CL baselines are originally proposed with CNN architecture. Meanwhile, we employ the contrastive learning objectives with mixing augmentations for BYOL and SimSiam proposed in BSIM (Chu et al., 2022) because these CL baselines adopt the MSE loss between positive sample pairs instead of the infoNCE loss (Eq. **??**).

**CL methods with Mixup augmentations.** In Sec. 4.2, we compare the proposed SAMix variants with general Mixup approaches proposed in SL and well-designed CL methods with Mixups. As for the general Mixup variants implemented with CL baselines, Mixup (Zhang et al., 2018), Manifold-Mix (Verma et al., 2019), CutMix (Yun et al., 2019), SaliencyMix (Uddin et al., 2021), FMix (Harris et al., 2020), ResizeMix (Qin et al., 2020), PuzzleMix (Kim et al., 2020), and out proposed SAMix only use the single Mixup augmentation. As for the CL methods applying Mixup augmentations, DACL (Verma et al., 2021) employs the vanilla Mixup, MoCHi (Kalantidis et al., 2020), i-Mix (Lee et al., 2021), UnMix (Shen et al., 2021), WBSIM (Chu et al., 2022) use two types of Mixup strategies in the input image and the latent space of the encoder, SDMP (Ren et al., 2022) randomly applies three types of input space Mixups (Mixup, CutMix, and ResizeMix). Therefore, SDMP can achieve competitive performances as SAMix variants in Table 7 and Table 8.

**Evaluation protocols.** We evaluate the SSL representation with a linear classification protocol proposed in MoCo (He et al., 2020b) and MoCo.V3 (Chen et al., 2021) for ResNet and ViT variants, which trains a linear classifier on top of the frozen representation on the training set. For ResNet variants, the linear classifier is trained 100 epochs by an SGD optimizer with the SGD momentum of 0.9 and the weight decay of 0. We set the initial learning rate of 30 for IN-1k as MoCo, and 0.1 for STL-10 and Tiny with a batch size of 256. The learning rate decays by 0.1 at epochs 60 and 80. For ViT-S, the linear classifier is trained 90 epochs by the SGD optimizer with a batch size of 1024 and a basic learning rate of 12. Moreover, we adopt object detection task to evaluate transfer learning abilities following MoCo, which uses the 4-th layer feature maps of ResNet (ResNet-C4) to fine-tune Faster R-CNN (Ren et al., 2015) with 24k iterations on the *trainval07+12* set and Mask R-CNN (He et al., 2017) with 2× training schedule (24-epoch) on the *train2017* set.

## A.5 EMPIRICAL EXPERIMENTS

### A.5.1 CROSS-VIEW TRAINING PIPELINE FOR INSTANCE-LEVEL MIXUP

We first analyze the learning objective of instance-level mixup classification for contrastive learning (CL) as discussed in Sec. 2. As shown in Figure 8 (a), there are two possible objectives for instance-level mixup defined in Eq. 2: the *same-view*, $\max_{\theta,\omega} \lambda I(z_m^{\tau_q}, z_i^{\tau_q}) + (1-\lambda)I(z_m^{\tau_q}, z_j^{\tau_q})$, and *cross-view* objective, $\max_{\theta,\omega} \lambda I(z_m^{\tau_q}, z_i^{\tau_k}) + (1-\lambda)I(z_m^{\tau_q}, z_j^{\tau_k})$. We hypothesize that the cross-view objective yields better CL performance than the same-view because the mutual information between two augmented views should be reduced while keeping task-relevant information (Tian et al., 2020b; Tsai

et al., 2021). To verify this hypothesis, we design an experiment of various mixup methods with $\alpha = 1$ on STL-10 with ResNet-18. As shown in Figure 8 (b), we compare using the same-view or cross-view pipelines combined with using $\ell^{NCE}(z_i^{\tau_q}, z_i^{\tau_k}) + \ell^{NCE}(z_m)$ or only using $\ell^{NCE}(z_m)$. We can conclude: (i) Degenerated solutions occur when using the same-view pipeline while using the cross-view pipeline outperforms the CL baseline. It is mainly caused by degenerated mixed samples which contain parts of the same view of two source images. Therefore, we propose the cross-view pipeline for the instance-level mixup, where $z_i$ and $z_j$ in Eq. 2 are representations of $x_i^{\tau_k}$ and $x_j^{\tau_k}$. (ii) Combining both the original and mixup infoNCE loss, $\ell^{NCE}(z_i^{\tau_q}, z_i^{\tau_k}) + \ell^{NCE}(z_m)$, surpasses only using one of them, which indicates that mixup enables $f_\theta$ to learn the relationship between local neighborhood systems.

### A.5.2 ANALYSIS OF INSTANCE-LEVEL MIXUP

As we discussed in Sec. A.5.1, we propose the cross-view training pipeline for instance-level mixup classification. We then discuss inter- and intra-class proprieties of instance-level mixup. As shown in Figure 8 (c), we adopt inter-cluster and intra-cluster mixup from {Mixup, CutMix} with $\alpha \in \{0.2, 1, 2, 4\}$ to verify that instance-level mixup should treat inter- and intra-class mixup differently. Empirically, mixed samples provided by Mixup preserve global information of both source samples (smoother), while samples generated by CutMix preserve local patches (more discriminative). And we introduce pseudo labels (PL) to indicate different clusters by clustering method ODC (Zhan et al., 2020) with the class (cluster) number $C$. Based on experiment results, we can conclude that inter-class mixup requires *discriminative* mixed samples with *strong* intensity while the intra-class needs *smooth* samples with *low* intensity. Moreover, we provide two cluster-based instance-level mixup methods in Table 5 and 6 (denoting by $*$): (a) Inter-Intra$^*$. We use CutMix with $\alpha \geq 2$ as inter-cluster mixup and Mixup with $\alpha = 0.2$ as an intra-cluster mixup. (b) PuzzleMix$^*$. We introduce saliency-based mixup methods to SSL tasks by introducing PL and a parametric cluster classifier $g_\psi^C$ after the encoder. This classifier $g_\psi^C$ and encoder $f_\theta$ are optimized online like AutoMix and SAMix mentioned in A.2. Based on Grad-CAM (Selvaraju et al., 2019) calculated from the classifier, PuzzleMix can be adopted on SSL tasks.

### A.5.3 ANALYSIS OF MIXUP GENERATION OBJECTIVES

In Sec. 3.1, we design experiments to analyze various losses for mixup generation in Figure 3 (left) and the proposed $\eta$-balanced loss in Figure 3 (right) for both SL and SSL tasks with ResNet-18 on STL-10 and Tiny. Basically, we assume both STL-10 and Tiny datasets have 200 classes on their 100k images. Since STL-10 does not provide ground truth labels (L) for 100k unlabeled data, we introduce PL generated by a supervised pertained classifier on Tiny as the "ground truth" for its 100k training set. Notice that L denotes ground truth labels, and PL denotes pseudo labels generated by ODC (Zhan et al., 2020) with $C = 200$.

As for the SL task, we use the labeled training set for mixup classification (100k on Tiny *v.s.* 5k on STL-10). Notice that SL results are worse than using SSL settings on STL-10, since the SL task only trains a randomly initialized classifier on 5k labeled data. Because the infoNCE and BCE loss require cross-view augmentation (or they will produce trivial solutions), we adopt MoCo.V2 augmentation settings for these two losses when performing the SL task. Compared to CE (L), we corrupt the global term in CE as CE (PL) or directly remove them as pBCE (L) to show that pBCE is vital to optimizing mixed samples. Similarly, we show that the global term is used as the global constraint by comparing BCE (UL) with infoNCE (UL), infoNCE (PL), and infoNCE (L).

As for the SSL task, we verify the conclusions drawn from the SL task and conclude that (a) the local term optimizes mixup generation directly, corresponding to the smoothness property, and (b) the global term serves as the global constraint corresponding to the discriminative property. Moreover, we verified that using the $\eta$-balanced loss as $\mathcal{L}_\phi^{cls}$ yields the best performance on SL and SSL tasks. Notice that we use $\eta = 0.5$ on small-scale datasets and $\eta = 0.1$ on large-scale datasets for SL tasks and use $\eta = 0.5$ for all SSL tasks.

### A.5.4 ANALYSIS OF MUTUAL INFORMATION FOR MIXUP

Since mutual information (MI) is usually adopted to analyze contrastive-based augmentations (Tian et al., 2020a;b), we estimate MI between $x_m$ of various methods and $x_i$ by MINE (Belghazi et al.,

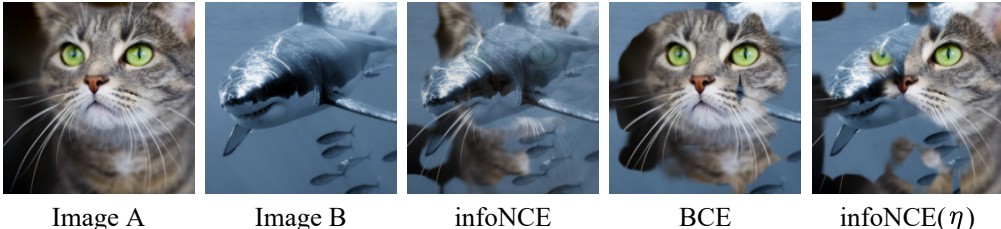

| Image A | Image B | infoNCE | BCE | infoNCE($\eta$) |

Figure 9: Visualization of loss effect. Both infoNCE and BCE loss have different emphases: infoNCE shows a similar effect of supervised fine-grained classification, focusing on fragmented and essential features, while BCE focuses on object completeness.

2018) with 100k images in 64×64 resolutions on Tiny-ImageNet. We sample $\lambda =$ from 0 to 1 with the step of 0.125 and plot results in Figure 7 (d). Here, we see that SAMix-C and SAMix-I with more MI when $\lambda \approx 0.5$ perform better.

### A.5.5 RESULTS OF DOWNSTREAM TASKS

In Sec. 4.2, we evaluate transferable abilities of the learned representation of self-supervised methods to object detection task on PASCAL VOC (Everingham et al., 2010) and COCO (Lin et al., 2014). In Table 11, the online SAMix-C and the pre-trained SAMix-I$^{\mathcal{P}}$ achieve the best detection performances among the compared methods and significantly improves the baseline MoCo.V2 (*e.g.,* SAMix-C gains 0.9% AP and +0.7% AP$^b$ over MoCo.V2). Notice that MoCHi, i-Mix, and UnMix introduce mixup augmentations in both the input and latent spaces, while our proposed SAMix only generates mixed samples in the input space.

Table 11: Transferring to object detection with Faster R-CNN on VOC and Mask R-CNN on COCO.

| CL Method | Methods | Faster R-CNN | | | Mask R-CNN | | |
|---|---|---|---|---|---|---|---|
| | | AP | AP$_{50}$ | AP$_{75}$ | AP$^b$ | AP$^b_{50}$ | AP$^b_{75}$ |
| MoCo.V2 | - | 56.9 | 82.2 | 63.4 | 40.6 | 60.1 | 44.0 |
| MoCo.V2 | Mixup | 57.4 | 82.5 | 64.0 | 41.0 | 60.8 | 44.3 |
| MoCo.V2 | CutMix | 57.3 | 82.7 | 64.1 | 41.1 | 60.8 | 44.4 |
| MoCo.V2 | Inter-Intra$^\star$ | 57.5 | 82.8 | 64.2 | 41.2 | 60.9 | 44.4 |
| MoCHi$^\dagger$ | *input+latent* | 57.1 | 82.7 | 64.1 | 41.0 | 60.8 | 44.5 |
| i-Mix$^\dagger$ | *input+latent* | 57.5 | 82.7 | 64.2 | - | - | - |
| UnMix$^\ddagger$ | *input+latent* | 57.7 | 83.0 | 64.3 | 41.2 | 60.9 | **44.7** |
| WBSIM$^\ddagger$ | *input* | 57.4 | 82.8 | 64.2 | 40.7 | 60.8 | 44.2 |
| **MoCo.V2** | **SAMix-I** | 57.5 | 83.1 | 64.2 | 41.2 | 61.0 | 44.5 |
| **MoCo.V2** | **SAMix-I$^{\mathcal{P}}$** | **57.8** | **83.2** | 64.3 | **41.3** | **61.1** | 44.6 |
| **MoCo.V2** | **SAMix-C** | 57.7 | 83.1 | **64.4** | **41.3** | **61.1** | **44.7** |

### A.6 VISUALIZATION OF SAMIX

#### A.6.1 MIXING ATTENTION AND CONTENT IN MIXER

In Sec. 3.2, we discuss the trivial solutions of Mixer, which usually occur in SSL tasks. Given the sample pair $(x_i, x_j)$ and $\lambda = 0.5$, we visualize the content $C_i$ and $P_{i,j}$ to compare the trivial and non-trivial results in the SSL task on STL-10, as shown in Figure 4. As we can see, both $C_i$ and $P_{i,j}$ from the trivial solutions have extremely large or small scale values while $C_i$ generated by $C_{NCL}$ containing more balanced values. Since the attention weight $P_{i,j}$ is normalized by softmax, we hypothesize that $C_i$ more likely causes trivial solutions. To verify our hypothesis, we freeze $W_P$ in the original MB and compare the original linear content projection $W_z$ with the non-linear content modeling. The results confirm that the non-linear module can prevent large-scale values on $C_i$ and eliminate the trivial solutions.

#### A.6.2 EFFECTS OF MIXUP GENERATION LOSS

In addition to Sec.3.3, we further provide visualization of mixed samples using the infoNCE (Eq. 2), BCE (Eq. 5), and $\eta$-balanced infoNCE loss (Eq. 6) for Mixer. As shown in Figure 9, we find that

mixed samples using infoNCE mixup loss prefer instance-specific and fine-grained features. On the contrary, mixed samples of the BCE loss seem only to consider discrimination between two corresponding neighborhood systems. It is more inclined to maintain the continuity of the whole object relative to infoNCE. Thus, combining both the characteristics, the $\eta$-balanced infoNCE loss yields mixed samples that retain both instance-specific features and global discrimination.

### A.6.3 VISUALIZATION OF MIXED SAMPLES IN SAMIX

**SAMix in various scenarios.** In addition to Sec. 3.3, we visualize the mixed samples of SAMix in various scenarios to show the relationship between mixed samples and class (cluster) information. Since IN-1k contains some samples in CUB and Aircraft, we choose the overlapped samples to visualize SAMix trained for the fine-grained SL task (CUB and Aircraft) and SSL tasks (SAMix-I and SAMix-C). As shown in Figure 10, mixed samples reflect the granularity of class information adopted in mixup training. Specifically, we find that mixed samples using infoNCE mixup loss (Eq.2) is more closely to the fine-grained SL because they both have many fine-grained centroids.

**Comparison with PuzzleMix in SL tasks.** To highlight the accurate mixup relationship modeling in SAMix compared to PuzzleMix (standing for saliency-based methods), we visualize the results of mixed samples from these two methods in the supervised case in Figure 11. There is three main difference: (a) bilinear upsampling strategy in SAMix makes the mixed samples smoother in local patches. (b) adaptive $\lambda$ encoding and mixing attention enhances the correspondence between mixed samples and $\lambda$ value. (c) $\eta$-balanced mixup loss enables SAMix to balance global discriminative and fine-grained features.

**Comparison of SAMix-I and SAMix-C in SSL tasks.** As shown in Figure 12, we provide more mixed samples of SAMix-I and SAMix-C in the SSL tasks to show that introducing class information by PL can help Mixer generate mixed samples that retain both the fine-grained features (instance discrimination) and whole targets.

### A.7 DETAILED RELATED WORK

**Contrastive Learning.** CL amplifies the potential of SSL by achieving significant improvements on classification (Chen et al., 2020a; He et al., 2020b; Caron et al., 2020), which maximizes similarities of positive pairs while minimizing similarities of negative pairs. To provide a global view of CL, MoCo (He et al., 2020b) proposes a memory-based framework with a large number of negative samples and model differentiation using the exponential moving average. SimCLR (Chen et al., 2020a) demonstrates a simple memory-free approach with large batch size and strong data augmentations that is also competitive in performance to memory-based methods. BYOL (Grill et al., 2020) and its variants (Chen & He, 2021; Chen et al., 2021) do not require negative pairs or a large batch size for the proposed pretext task, which tries to estimate latent representations from the same instance. Besides pairwise contrasting, SwAV (Caron et al., 2020) performs online clustering while

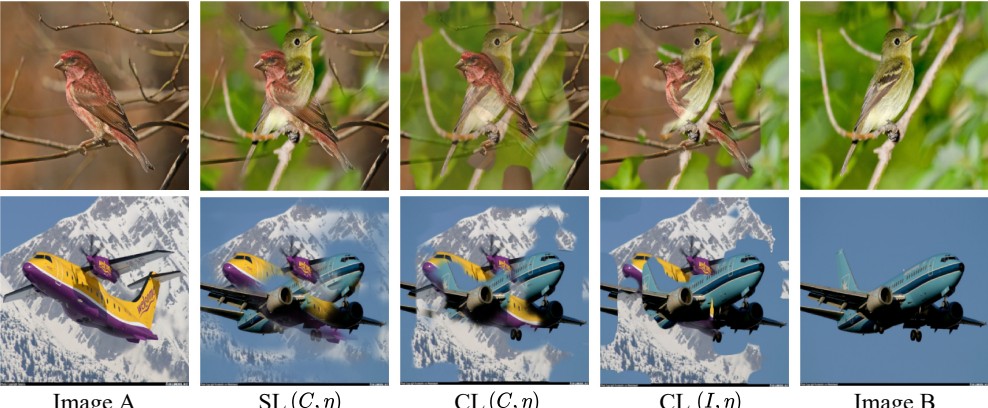

| Image A | SL $(C, \eta)$ | CL $(C, \eta)$ | CL $(I, \eta)$ | Image B |

Figure 10: Visualization of SAMix in various scenarios on CUB and Aircraft. Given images A and B, the middle three mixed samples are generated by SAMix with $\lambda = 0.5$ trained in the fine-grained SL task and the SSL tasks (SAMix-C and SAMix-I).

enforcing consistency between multi-views of the same image. Barlow Twins (Zbontar et al., 2021) avoids the representation collapsing by learning the cross-correlation matrix of distorted views of the same sample. Moreover, MoCo.V3 (Chen et al., 2021) and DINO (Caron et al., 2021) are proposed to tackle unstable issues and degenerated performances of CL based on popular Vision Transformers (Dosovitskiy et al., 2021).

**Mixup.** MixUp (Zhang et al., 2018), convex interpolations of any two samples and their unique one-hot labels were presented as the first mixing-based data augmentation approach for regularising the training of networks. ManifoldMix (Verma et al., 2019) and PatchUp (Faramarzi et al., 2020) expand it to the hidden space. CutMix (Yun et al., 2019) suggests a mixing strategy based on the patch of the image, *i.e.*, randomly replacing a local rectangular section in images. Based on CutMix, ResizeMix (Qin et al., 2020) inserts a whole image into a local rectangular area of another image after scaling down. FMix (Harris et al., 2020) converts the image to Fourier space (spectrum domain) to create binary masks. To generate more semantic virtual samples, offline optimization algorithms are introduced for the saliency regions. SaliencyMix (Uddin et al., 2021) obtains the saliency using a universal saliency detector. With optimization transportation, PuzzleMix (Kim et al., 2020) and Co-Mixup (Kim et al., 2021) present more precise methods for finding appropriate mixup masks based on saliency statistics. SuperMix (Dabouei et al., 2021) combines mixup with knowledge distillation, which learns a pixel-wise sample mixing policy via a teacher-student framework. More recently, TransMix (Chen et al., 2022) and TokenMix (Liu et al., 2022a) are proposed specially designed Mixup augmentations for Vision Transformers (Dosovitskiy et al., 2021). Differing from previous methods, AutoMix (Liu et al., 2022b) can learn the mixup generation by a sub-network end-to-end, which generates mixed samples via feature maps and the mixing ratio. Orthogonal to the sample mixing strategies, some researchers Park et al. (2022); Chen et al. (2022); Liu et al. (2023) improve the label mixing policies upon linear mixup.

**Mixup for contrastive learning.** A complementary method for better instance-level representation learning is to use mixup on CL (Kalantidis et al., 2020; Shen et al., 2021). When used in collaboration with CE loss, Mixup and its several variants provide highly efficient data augmentation for SL by establishing a relationship between samples. Most approaches are limited to linear mixup methods without a ground-truth label. For example, Un-mix (Shen et al., 2021) attempts to use MixUp in the input space for self-supervised learning, whereas the developers of MoChi (Kalantidis et al., 2020) propose mixing the negative sample in the embedding space to increase the number of hard negatives but at the expense of classification accuracy. i-Mix (Lee et al., 2021), DACL (Verma et al., 2021), BSIM (Chu et al., 2022) and SDMP (Ren et al., 2022) demonstrated how to regularize contrastive learning by mixing instances in the input or latent spaces. We introduce an automatic mixup for SSL tasks, which adaptively learns the instance relationship based on inter- and intra-cluster properties online.

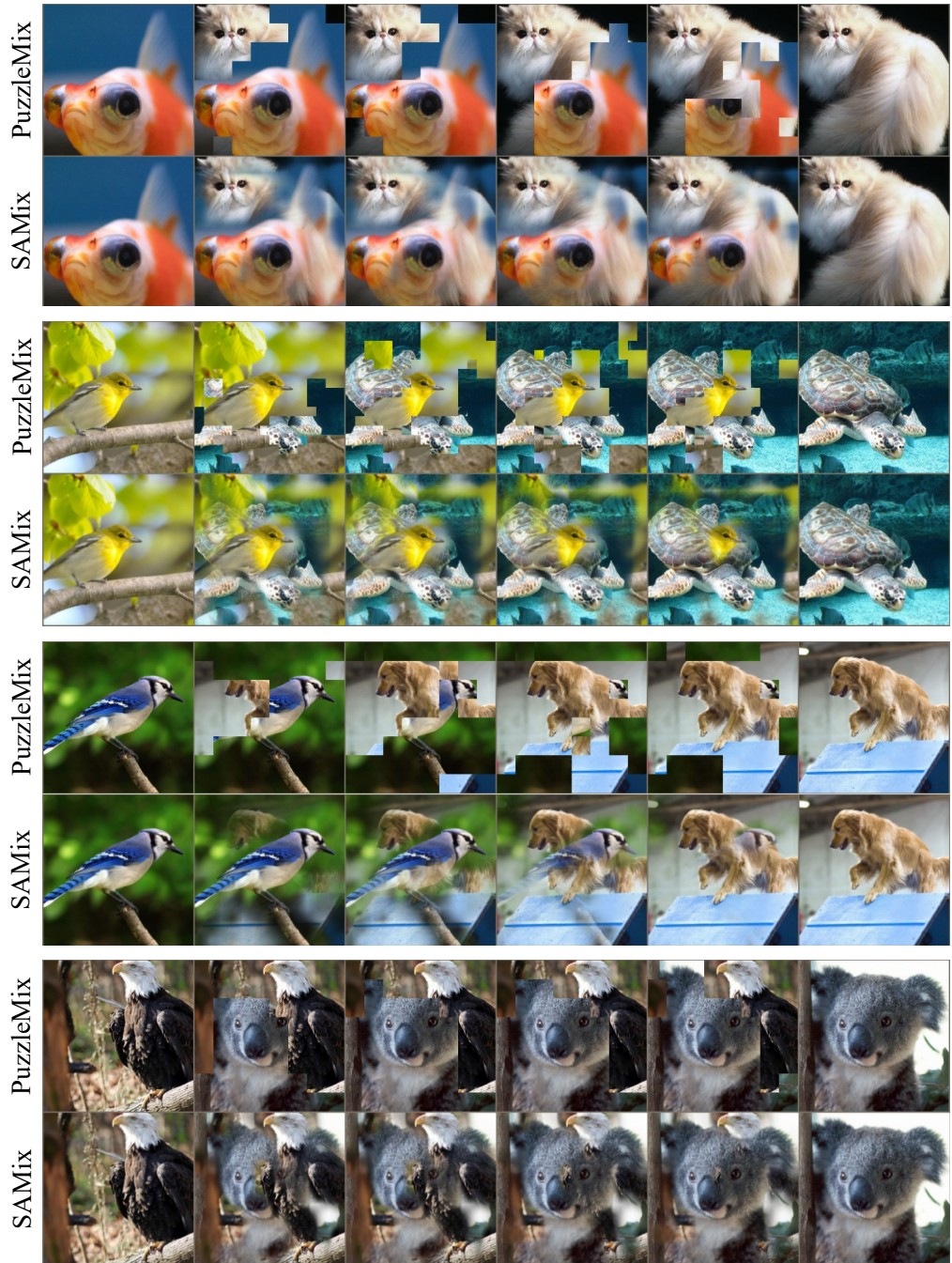

Figure 11: Visualization of PuzzleMix *v.s.* SAMix for SL tasks on IN-1k. In every four rows, the upper and lower two rows represent mixed samples generated by PuzzleMix and SAMix, respectively. $\lambda$ value changes from left ($\lambda = 0$) to right ($\lambda = 1$) by an equal step.

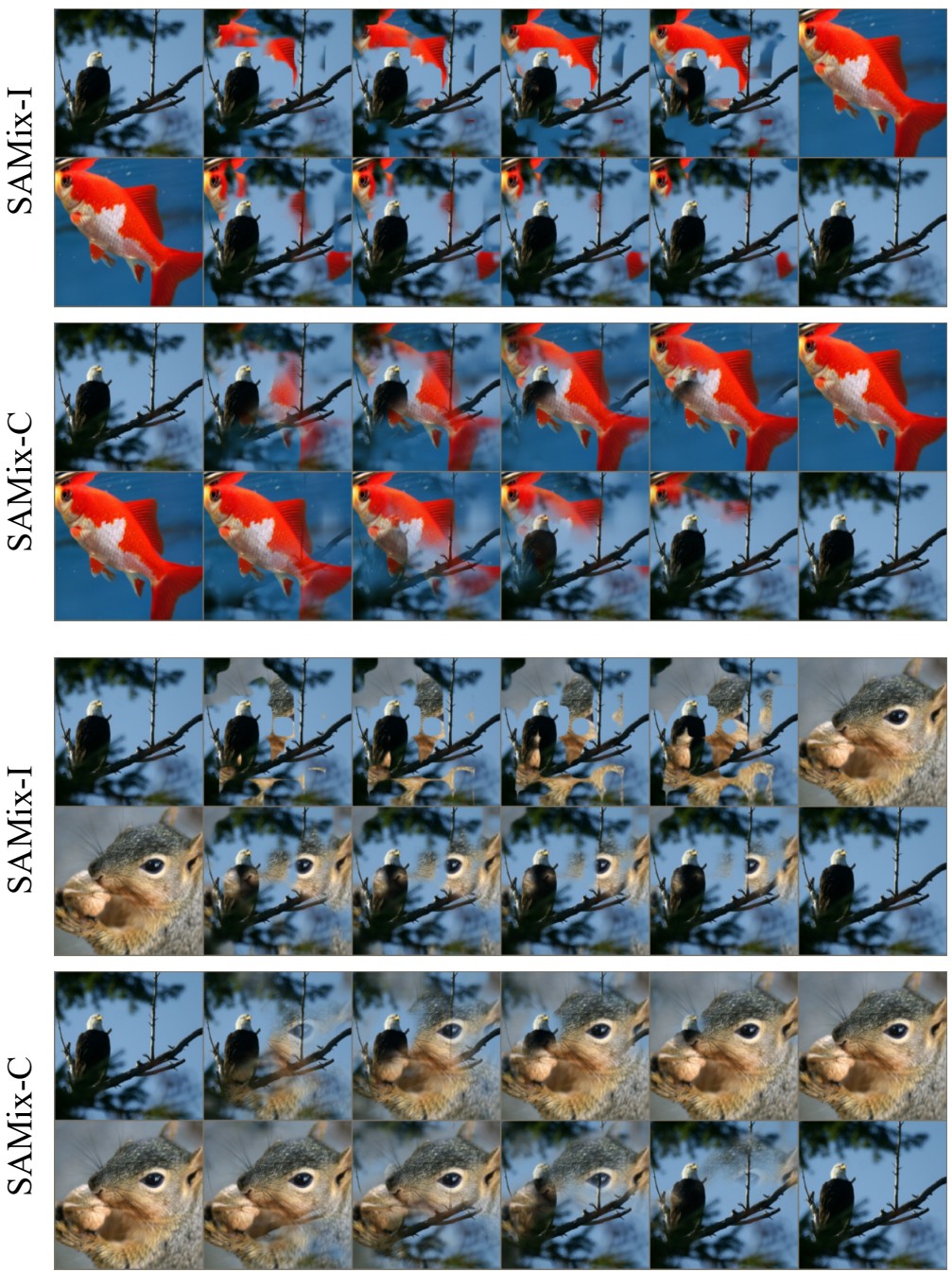

Figure 12: Visualization of SAMix-I *v.s.* SAMix-C for SSL tasks on IN-1k. In every four rows, the upper and lower two rows represent mixed samples generated by SAMix-I and SAMix-C, respectively. $\lambda$ value changes from left ($\lambda = 0$) to right ($\lambda = 1$) by an equal step.

