# OpenReview forum: "Boosting Discriminative Visual Representation Learning with Scenario-Agnostic Mixup"
_ICLR.cc/2024/Conference — ICLR 2024 Conference Withdrawn Submission_

### Official Review · Reviewer_3BML · 2023-10-25

**Soundness:** 2 fair
**Presentation:** 3 good
**Contribution:** 2 fair
**Rating:** 6
**Confidence:** 4

**Summary:**

The paper studies mixup generation and introduces Scenario-Agnostic Mixup (SAMix) to address two major challenges: performance variation across scenarios due to trivial solutions, and the Self-supervised learning (SSL) dilemma in online training policies. The authors propose an η-balanced mixup loss for balanced learning and introduce a label-free generator for non-trivial sample mixing. A pre-trained version, SAMixP, is also provided to reduce computational costs and improve generalizability.

**Strengths:**

• The research topic is important.  Mixup is an impoartant and useful data augmentation in reality.

• The idea of introducing a label-free generator for non-trivial sample mixing is novel.

• The experiment details are clear enough for reproducibility. The writing is in general good.

**Weaknesses:**

• In equation1, what does $P_m$ stands for ?

• SAMix proposes  a versatile mixup framework with η-balanced mixup loss and a specialized Mixer and provide two version of learning objective , SAMix-C and SAMix-I , then came up with the final  η-balanced mixup loss objective, but how can one be sure to choose the value of η ?

• The introduction motivates the paper that SAMix has a  a robust sample mixing policy but the author  do not provide any experiment or theoretical proof for their claim.

**Questions:**

1. Clarify the notations in all formulation such as different training objective, and better to have a separate algorithm workflow for training under Mixup classification and Mixup generation scenarios.

2. The authors mentioned that $SAMix^P$ exhibits better transferabilities than the popular $AutoMix^P$, but they only compare $SAMix^P$ with $AutoMix$, are there any reasons for choosing the comparison baseline?

3. Furthermore, it remains unclear whether the performance gains are significant, since no standard deviations of the performance metrics are provided. So, better to add standard deviations for performance comparison.

4. Better to testify SAMix on some corrupted dataset to justify the robustness of Mixup Policy.

---

### Official Review · Reviewer_2ZAN · 2023-10-31

**Soundness:** 1 poor
**Presentation:** 1 poor
**Contribution:** 2 fair
**Rating:** 3
**Confidence:** 2

**Summary:**

This paper considers the problem of mixup generation and classification in a discriminative-contrastive setting where the discriminative part considers mixing samples from different classes while the contrastive part considers instance level mixup. The paper argues that such a setup can be treated as a optimizing for local smoothness versus global discrimination, respectively; however has two shortcomings: i) it is prone to trivial solutions and ii) lower empirical performances in self-supervised learning situations. To this end, the paper suggests scenario-agnostic mix (SAMix) that implements three ideas: a) learns the mixing ratio \lambda, b) avoids trivial solutions via non-linear content modeling (eq. 9), and c) expensive compute for online training via using pre-trained models. Experiments are provided on 12 tasks and demonstrate some promise.

**Strengths:**

The paper is very poorly written and thus it is hard for me to judge the technical strength or its contributions. However, the empirical evidence reported in the paper shows some promise in some situations. For example, while the performance improvements are very minor (less than 0.5%) in Tables 1,2,3,4, when using pre-trained models (Tables 5,6,7,8) they are slightly higher (1-2%).

**Weaknesses:**

The paper needs a thorough revision for clarity and technical correctness, and preciseness in the assumptions it is making. Some suggestions are provided below to consider when revising.
1. It is unclear to me what exactly is meant by "scenario" in the paper? Is it class or instances? Where exactly is the "agnostic" part coming in?
2. The main hypothesis in the paper is that mixup generation can be treated as a sum of local and global terms (Sec. 3.1). But it appears that which one is local and which is global is not quite clear from the exposition. For example, the paper writes "the local term centers around classes, while global term is across classes". Then writes "\ell^CE is a global term" and then writes "at the class level, to emphasize the local term, we introduce binary cross-entropy loss pBCE". Further, in Eq. 4, the paper assume access to the labels y_i and y_j, and so does for the NCE loss in Eq. 5, which is supposed to be label free (as described earlier in the Preliminaries)?

3. In Sec 3.2, Eq. 7 is claimed to be an adaptive mixing factor \lambda, but it is not about updating lambda. Further, the findings in Sec 3.2, which is essentially the main contributions of the paper, the observations are only empirical and the insights are not substantiated in any rigorous manner. The fixes, such as adaptive \lambda, non-linear content modeling, etc. are only treated superficially and it is not clear to me why it makes sense or how is the particular formulation justified. For example, the paper writes "it may be unstable to project high-dimensional features to 1-dim linearly", which the paper claims to be the reason for trivial solutions, and the offering is to use two 1x1 convolution layers with a batch regularization and ReLU.


Other comments:
a. f_\theta(x): x->z. Typically, x and z are replaced by their domains not their variables. Also, the domain of z is not defined. Is it d_c? Do you assume each class has a separate dimension?

b. "closed-loop framework": how is it closed loop?

c. Please double check Eq. 1; it is incorrect. The lambdas should go inside \ell^CE I believe.

d. Why do you use the same \lambda for \ell^NCE and \ell^CE (e.g. in eq.1 and eq. 2), when they look at instance level and class level losses? For that matter, what precisely is meant by parametric loss and non-parametric loss for \ell^CE  and \ell^NCE, respectively? When adding these two losses in the same objective (e.g., Eq. 3), how do you ensure it is optimized correctly, as NCE loss I believe do not assume knowledge of the class labels, while CE loss discriminates classes? The term L^{cls} is undefined. What is the precise form for M_\phi, and how is z_i, z_j used?

e. "SSL framework can easily fall into a trivial solution". This is a strong statement and is mostly an empirical observation. It would useful to provide a more technical insight to the reason for this.

Overall, this paper needs a thorough revision for language and technical clarity. As is, the contributions appear as minor hacks to improve performance.

**Questions:**

Please see above.

**Details Of Ethics Concerns:**

There is no concern of ethical issues in this paper as far as I see it.

---

### Official Review · Reviewer_1gXV · 2023-11-02

**Soundness:** 2 fair
**Presentation:** 1 poor
**Contribution:** 2 fair
**Rating:** 5
**Confidence:** 4

**Summary:**

- The paper analyses mixup by breaking it into two subtasks : of generating a mixed example and classifying it.
- A loss function is proposed which balances local smoothness (characteristic of mixup) and global discrimination to other classes.
- The authors propose a learnable module to avoid trivial solutions and perform optimal mixup.
- A pretrained version of the module - SAMix^{P} is also proposed which performs comparably while relaxing training computational requirements.

**Strengths:**

- A general approach which seems to work for both supervised and self-supervised approaches.
- The overall approach is quite complicated with many moving parts, the use of a pretrained mixer can help in use with other datasets with small overheads.
- The approach has been extensively tested and analyzed. A detailed study like this will be helpful to the community (if the clarity issues are resolved).

**Weaknesses:**

Approach, Experiments:
- The improvements over previous best seem to be marginal in most cases.
- The training time is an interesting tradeoff - while it seems like there is performance improvement, it often comes with a significant increase in training time. Can the authors provide a similar analysis on time for SSL experiments ?

Clarity and presentation:
- There are many small (potential) typos, grammatical errors or sentences which should be rephrased and are not apt for a professional paper. The paper needs a thorough rewrite to fix these issues. That said, I consider most of these as minor issues and do not play a big role in my current score.
- The more pressing concern is that the paper has numerous clarity concerns. Even distilling the main crux of the sentence is difficult at times. Many sentences are "hand-wavy". Some examples:
    - Section 1:
        - "they still do not ... scenarios" - too convoluted. This is a problem since this is likely the most important sentence to motivate the introduction (and the idea). No intuition on trivial solutions.
        - [Minor] Figure 1 : Too many variants make this difficult to interpret. Make sure that fig1 and fig2 caption are easy to demarcate.
    - Section 2:
        - "Here we consider .. framework". Unclear.
        - "Symmetrically" has been used too loosely in the entire paper.
        - The paragraph between (1) and (2) suddenly switches to InfoNCE. Separate these.
        - Notation Page (3) : $z_m$ not defined, define "augmentation view". Unclear ".. not from the same view.."
        - "unlike .. in Sec 2" - this sentence is still a part of Sec 2.
        - Abrupt change from $z_m$ back to discussing $x_m$
        - "depends on quality of mixup generation" - motivation unclear till this point. All I get from this para is that you want to train a model to mix. What are we missing by using heuristics to generate $x_m$ ?
        - What is the difference between $\mathcal{L}^{gen}$ and $\mathcal{L}^{phi}$ and $\mathcal{L}^{cls}$
    - Section 3:
       - "Typically .. parametric training" - what does this mean ?
       - Intuitions for (4) and (5) ? What does the proposed BCE loss do ? What are the variables being optimized ?
       - Again, the section on "balancing local and global.." is not well written, intuitions are not clear. I suggest shortening this and moving empirical results from this to the experiments section to improve flow of the paper. Since PL forms an integral part of your paper, provide some details/intuitions instead of just referring to a prior work.
       - Cite 'AutoMIx' when you talk about it in 3.2

[Minor] Some Missing references which rely on mixup-like ideas for self-supervised learning [a,b]


[a] HaLP: Hallucinating Latent Positives for Skeleton-based Self-Supervised Learning of Actions (CVPR'23)

[b] Hallucination Improves the Performance of Unsupervised Visual Representation Learning (ICCV'23)

**Questions:**

Please refer to the weaknesses section. Other miscellaneous questions:
- Is the approach relevant for the recent masked image modeling based SSL methods ?
- What was the overhead over no-mixup when obtaining results like in Table 7 ?

**Details Of Ethics Concerns:**

No ethics concern

---

### Official Review · Reviewer_AzRU · 2023-11-04

**Soundness:** 2 fair
**Presentation:** 1 poor
**Contribution:** 2 fair
**Rating:** 5
**Confidence:** 3

**Summary:**

This paper proposes a variant of mixup called Scenario Agnostic Mixup (SAMix). Mixup generation is considered as an auxiliary task. The mixup generation sub-network called Mixer is trained in along with the encoder.

The mixup generation loss is divided into two components balanced by $\eta$.

The overall loss is the sum of classification/SSL loss and mixup generation loss, and both are optimized alternatively. EMA version of the encoder is used when optimizing the Mixer parameters.

Experiments are performed on 10 supervised learning datasets and 2 SSL datasets achieving SOTA results.

**Strengths:**

- A mixup method and objective which is applicable in both supervised (SL) and self-supervised (SSL) settings
- Extensive experimental results on 12 benchmarks (including SL and SSL datasets)
- Thorough ablations on the involved hyperparameters

**Weaknesses:**

- The paper is poorly written
- Major claims and hypothesis in the paper are not clearly motivated/difficult to follow, e.g. enforcing local smoothness and global discrimination
- More recent SSL methods such as Masked Autoencoders are missing from the SSL evaluation
- The method does not help in the SSL linear evaluation using the MOCOv3 method (which is the best performing among the SSL methods considered)

**Questions:**

- See weaknesses above